

# Population dynamics and seasonal migration patterns of *Spodoptera exigua* in northern China based on 11 years of monitoring data

Hao-Tian Ma[1,*], Li-Hong Zhou[2,*], Hao Tan[1], Xian-Zhi Xiu[1], Jin-Yang Wang[1] and Xing-Ya Wang[1]

[1] Shenyang Agricultural University, College of Plant Protection, Shenyang, Liaoning, China
[2] Liaoning Academy of Agricultural Sciences, Institute of Flower, Shenyang, Liaoning, China
[*] These authors contributed equally to this work.

Corresponding author
Xing-Ya Wang,
wangxingya20081@syau.edu.cn

## ABSTRACT

**Background**. The beet armyworm, *Spodoptera exigua* (Hübner), is an important agricultural pest worldwide that has caused serious economic losses in the main crop-producing areas of China. To effectively monitor and control this pest, it is crucial to investigate its population dynamics and seasonal migration patterns in northern China.
**Methods**. In this study, we monitored the population dynamics of *S. exigua* using sex pheromone traps in Shenyang, Liaoning Province from 2012 to 2022, combining these data with amigration trajectory simulation approach and synoptic weather analysis.
**Results**. There were significant interannual and seasonal variations in the capture number of *S. exigua,* and the total number of *S. exigua* exceeded 2,000 individuals in 2018 and 2020. The highest and lowest numbers of *S. exigua* were trapped in September and May, accounting for 34.65% ± 6.81% and 0.11% ± 0.04% of the annual totals, respectively. The average occurrence period was 140.9 ± 9.34 days during 2012–2022. In addition, the biomass of *S. exigua* also increased significantly during these years. The simulated seasonal migration trajectories also revealed varying source regions in different months, primarily originated from Northeast China and East China. These unique insights into the migration patterns of *S. exigua* will contribute to a deeper understanding of its occurrence in northern China and provide a theoretical basis for regional monitoring, early warning, and the development of effective management strategies for long-range migratory pests.

## INTRODUCTION

Migration is a key process in the ecology and population dynamics of many insect species. Many insects engage in seasonal migration to escape deteriorating environments, utilize suitable habitat resources for reproduction, or evade competition, predation, and parasitism (*Broquet & Peti, 2009*; *Chapman, Reynolds & Wilson, 2015*; *Wang et al., 2022*). In general, a variety of insects migrate from wintering areas at lower latitudes to higher latitudes in

spring to exploit habitat resources suitable for breeding at higher latitudes in summer (*Chapman, Reynolds & Wilson, 2015*; *Hu et al., 2016a*; *Hu et al., 2016b*). Nearly 3.5 trillion insects traverse the southern United Kingdom every year, transferring more than 3,000 tons of biomass (*Hu et al., 2016a*; *Hu et al., 2016b*). Therefore, insects potentially have great importance in the flow of ecological resources and provide critical ecosystem services globally (*Semmens et al., 2011*; *Yang & Gratton, 2014*; *Menz et al., 2019*; *Satterfield et al., 2020*; *Chowdhury et al., 2021a*; *Chowdhury et al., 2021b*). Moreover, the long-distance migration behavior of insects can also can also lead to sudden outbreaks and epidemics of plant diseases (*Wang et al., 2022*). At present, many studies have focused on a limited number of species, such as butterflies (*Meitner, Brower & Davis, 2004*; *Brower, Fink & Walford, 2006*; *Satterfield et al., 2020*; *Chowdhury et al., 2021a*; *Chowdhury et al., 2021b*; *Chowdhury et al., 2022*), and 3.29% of butterflies have been identified as migratory species (*Chowdhury et al., 2021a*; *Chowdhury et al., 2021b*). However, the migration behavior and population dynamics of the majority of species remain unknown (*Chowdhury et al., 2022*). Therefore, a better understanding of the migratory behavior of pests will help to clarify the core ecological characteristics and their evolution and ultimately improve the integrated control of migratory pests.

The beet armyworm, *Spodoptera exigua* (Hübner), is one of the most polyphagous migratory pests worldwide (*Feng et al., 2003*; *Guo, Wu & Wan, 2010*). It attacks more than 150 host plant species, including vegetables, flowers, grasses, weeds and many other crops (*Burris et al., 1994*). This pest originated in South Asia and caused severe outbreaks in Asia, Europe, Africa and North America (*Feng et al., 2003*; *Ehler, 2004*; *Zheng et al., 2009*). In China, *S. exigua* was first recorded in Beijing in 1892. In the 1980s, the areas damaged by *S. exigua* expanded rapidly, and *S. exigua* has spread throughout the main crop-producing areas and caused severe economic losses in China. For example, this pest spread through vegetable production in Tianjin in the ten years (1993–2003), infested a total area of more than 8,000 hm² and reduced annual welsh onion production by 30% (*Zheng et al., 2009*). In 1997, more than 2.667 million hm² of crops were seriously damaged by *S. exigua* in Shandong, Henan, Anhui and Hebei (*Luo, Cao & Jiang, 2000*). This moth species damaged vegetables in Yichun, Heilongjiang Province in 2012. In 2021, the moth was also found to be harmful to crops such as sweet potato, cotton, winter melon and other crops in Shihezi, Xinjiang (*Wang et al., 2022*). In general, the first-instar and second-instar larvae of *S. exigua* gregarious feed on the abaxial leaf surface, and the third-instar larvae begin to feed solitarily. Fourth-instar larvae of *S. exigua* enter the binge eating stage and begin to feed on large quantities of leaves and pods (*Zheng et al., 2009*). Adults can reproduce continuously in areas where the average temperature in January exceeds 12 °C and overwinter between 4 °C and 12 °C in January isotherms (*Jiang & Luo, 2010*). Currently, chemical insecticides are still the main strategy for controlling *S. exigua* in subtropical and temperate regions. However, *S.exigua* has a long history of exposure to a broad array of insecticides and has developed resistance to many of these insecticides, such as organophosphorus, carbamate, and pyrethroids (*Chaufaux & Ferron, 1986*).

In recent years, various methods have been employed to determine the migratory behavior of insects, including mark-release-recapture, light capture, insect radar, isotope

tracing, natural marker, flight chamber, pollen analysis, suction trap and trajectory analysis (*Chapman, Reynolds & Wilson, 2015*; *Westbrook et al., 2016*; *Zhang et al., 2016*; *Fu et al., 2017*; *Wu et al., 2018*; *Minter et al., 2018*; *Reich et al., 2021*; *Wang et al., 2023*). In particular, great progress has been made in the application of pheromones to control moths (*Tumlinson, Mitchell & Sonnet, 1981*; *Mitchell, Sugie & Tumlinson, 1983*). *S. exigua* has a strong migratory ability. This pest was first introduced to Hawaii in the 1880s and has since spread to many states in the United States. In less than 50 years, the pest has spread from Oregon to Florida and even migrated south from Mexico, reaching Central America and the Caribbean (*Mitchell, Sugie & Tumlinson, 1983*). By analysing data obtained from light trapping and weather maps from 1947 to 1966, it has been concluded that the population sources of *S. exigua* in the British Isles can be traced back to North Africa, the Canary Islands, Spain, and Portugal. Moreover, the peak occurrence for this moth pest in the Netherlands coincided with that in the UK, indicating that both countries share the same source of migratory insect. The migration of *S. exigua* has also been observed in other European countries, such as Norway, Switzerland, Finland, and Denmark. In these cases, it is possible that *S. exigua* originated from southern Moscow between Kursk and the Caspian Sea (*Mikkola, 1970*). Based on the monsoon route and the distribution characteristics of *S. exigua* in the U.S., a possible migration path in autumn has been proposed. This findings suggested that the beet armyworm migrates to southern wintering areas, such as California, as well as perennial regions (*e.g.*, Arizona and Florida) (*Ehler, 2004*). Furthermore, it is speculated that *S. exigua* migrates northwards to northern regions during late spring and early summer and returns to southern regions from higher latitudes in autumn based on observations of the round-trip migrations of other species in the Noctuidae family. In China, the migratory behavior of *S. exigua* has been observed in Fengxian County, Jiangsu Province, through analyses of black light trapping and ovary anatomy (*Han et al., 2004*).

Unlike most migratory insects, *S. exigua* does not exhibit an "oogenesis and flight antagonism syndrome" characterized by alternating flight and reproduction antagonism (*Jiang & Luo, 2010*). Previous studies have indicated that *S. exigua* migrates seasonally once a year in eastern China. At the turn of spring and summer, the overwintering adult moths migrate northwards in South China and the Yangtze River Basin and migrate back from mid-late August to mid-early September. In late September and mid-October, adults continue to migrate south to overwintering areas (*Si et al., 2012*). However, compared those of other migratory pests, such as *Mythimna separata* and *Loxostege sticticalis*, the migration routes of *S. exigua* in different ecological regions are still unclear. Therefore, monitoring the migration pattern of this pest and predicting its migration path can assist in identifying the source area of the insects, and formulating effective pesticide application strategies.

The aim of this study were to analyze the interannual and seasonal population dynamics and migration patterns of *S. exigua* based on 11 years of monitoring data collected using sex pheromone traps in Liaoning Province and to evaluate the possible migration sources of this species. The results of this study will contribute to a comprehensive evaluation of the migration behavior of *S. exigua* and serve as a foundation for further advancements in early warning technology and integrated pest management strategies.

## MATERIALS & METHODS

### Sex pheromone trap monitoring and sampling

Three sex pheromone traps (YFCB-IV; Pherobio Technology Co., Ltd., Beijing, China) were set up in a Welsh onion field (123.57°N, 41.82°E) at Liaoning Academy of Agricultural Sciences, in Shenyang, Liaoning Province, Northeast China, from May to November annually between 2012 and 2022 (Fig. 1). The traps used were cylindrical plastic devices, measuring approximately 30 cm in height and 18 cm in diameter, featuring 16 one-way entrances located on the top for capturing pests. The pheromones utilized in this study were small PVC lures (Pherobio Technology Co., Ltd., Beijing, China). The key components of these lures were cis-9, trans-12-tetradecadienyl acetate and cis-9-tetradecadienol. To maintain their trapping effectiveness, the pheromone lures were replaced every two weeks. Adult moths were collected from the trap bag beneath the trap every week, and the number of individuals was recorded. Field experiments were approved by the Chinese Academy of Agricultural Sciences (201003025), and none of the field surveys in the study involved endangered or protected species. All samples were preserved at −20 °C and subsequently stored at the Plant Protection College, Shenyang Agricultural University, Shenyang, Liaoning Province.

To elucidate the peak of *S. exigua* migration events, the categorization methodology introduced by *Lin, Sun & Chen (1963)* was adopted and defined based on the following set of criteria. These criteria were applied to each instance of capture, denoted as N. A week, denoted as week-m, was identified as a peak week if the moth capture in that week, $N_{week-m}$, was at least 10 and twice the number of the previous week's capture, $N_{week-(m-1)}$. If $N_{week-m}$ was a peak week and the following week's capture, $N_{week-(m+1)}$, was 10 or more, then the following week, week-(m+1), was also considered a peak week. However, if the capture in a particular week, $N_{week-m}$, was less than 10, that week, week-m, was classified as a non-peak period. Furthermore, each peak week or a continuous peak period separated by more than 2 non-peak weeks was considered the beginning of the next peak migration event.

### Trajectory simulation

To evaluate the possible migration routes of *S. exigua*, the Hybrid Single-Particle Lagrangian Integrated Trajectory (HYSPLIT) trajectory simulation model was used to calculate the backwards trajectories from the estimated migration regions (*Draxler & Rolph, 2013*) based on per 7-days intervals of data. This pest typically migrates for up to 36 hrs and stratified flight heights are typically in the range of 300–700 m (*Trumble & Baker, 1984*; *Ruberson et al., 1994*; *Jiang et al., 2002*; *Fu et al., 2017*). The model parameters were set as follows (*Lu, Zhai & Hu, 2013*): First, the monitoring point was set as the starting point for the reverse simulation, with 06:00 BST as the start time and a duration of 12/24 h for each simulation. Second, the flight altitudes for each simulation were 300 m, 500 m and 700 m above ground level. Third, all individual days of the week during the peak period were simulated separately and counted as a percentage. Finally, the endpoint of the trajectory in the sea was not valid. In this study, trajectory routes were analysed by the potential source contribution function (PSCF) analysis using MeteoInfo (*Wang, 2014*) to further analyse

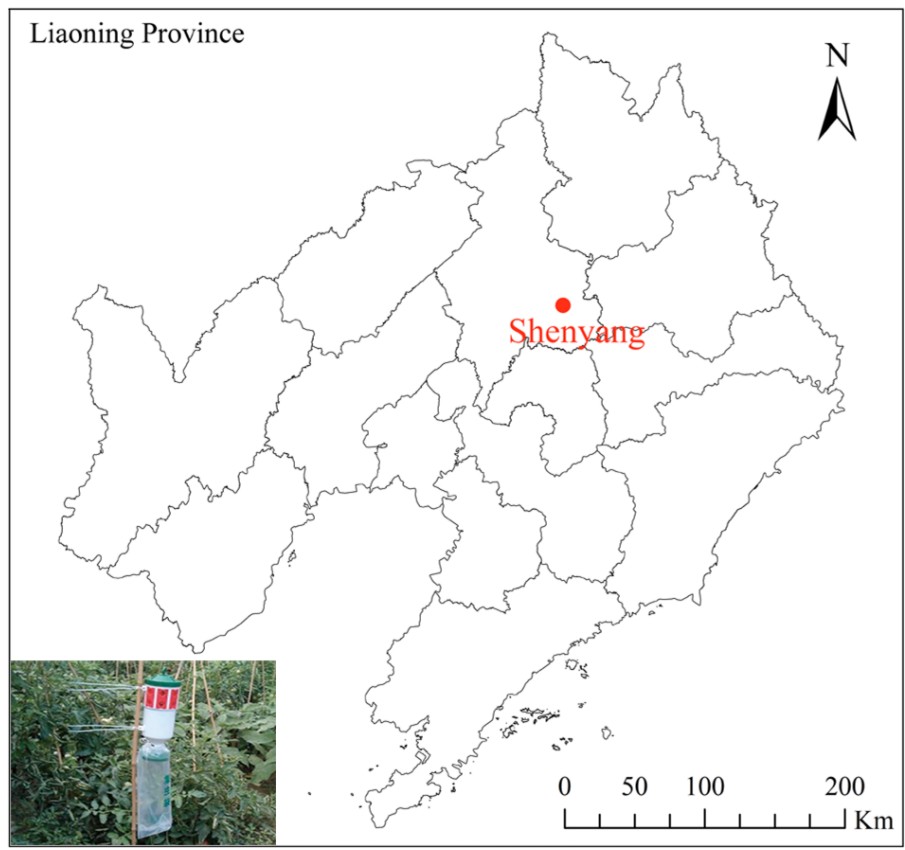

**Figure 1** **Map of the position of Shengyang and sex pheromone traps in northern China.** ArcGIS Pro (https://www.esri.com/en-us/arcgis/products/arcgis-pro/overview) was used to produce a monitoring and sampling position map based on the geographic coordinates. The base map (China: 1:1,000,000) for the analysis was obtained from the standard map. Map credit: GADM database (https://gadm.org/) for free use; the figure is modified from the graphic of *Zhu et al. (2020)*.

simulated migratory routes and count the number and percentage of migratory trajectory endpoints, marking all possible endpoints as areas where *S. exigua* could occur.

## Statistical analyses
### *Population dynamics of S. exigua*
Prior to the analysis, the Shapiro–Wilk test was used to check the normality of all the data, and the Levene test was used to check the homogeneity of variance. After logarithmic transformation of the number of traps in different years, a general linear model was used to analyse the interannual and seasonal differences of *S. exigua* from 2012 to 2021. If the variance analysis (ANOVA) showed a significant difference, SPSS 25.0 was used for Tukey's honest significant difference (Tukey's HSD) test (IBM, Armonk, NY, USA).

### *Interannual and seasonal migration patterns of S. exigua*
We utilized one-way ANOVA, Duncan's multivariate range test, and multiple paired *t*-tests to analyze the differences between the annual and seasonal occurrence numbers of *S. exigua*. Canoco version 4.5 (*Braak & Smilauer, 2002*) was used to analyze the effects of

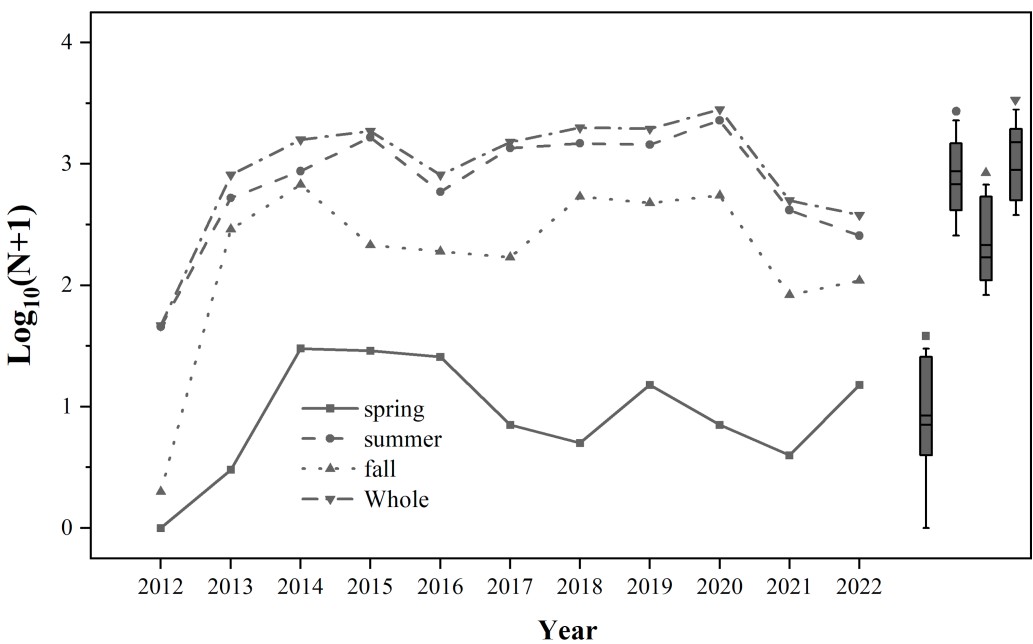

**Figure 2** **Population dynamics of trapped *Spodoptera exigua* in different seasons during 2012–2022.**
The means and medians of *Spodoptera exigua* caught in different seasons over the 11-year period are represented by the two short lines in the box plots.

various meteorological factors on the capture of *S. exigua*. First, detrended correspondence analysis was performed on the data, and the best sorting method was selected based on the length of the gradient axis. Second, we performed a quantitative analysis of meteorological factors using a backward selection and Monte Carlo arrangement tests to determine the interpretability of meteorological factors for trapping important migratory pests and dominant natural enemies by the use of sex pheromones, and the significance of meteorological variables was also tested.

## RESULTS

### Population dynamics of *S. exigua* using sex pheromone traps

From 2012 to 2022, the number of trapped *S. exigua* was the highest in summer, followed by autumn, and the lowest in spring (Fig. 2). A total of 14,095 individuals of *S. exigua* were captured, with an average annual catch of $1,281.4 \pm 260.5$ individuals. The highest number of *S. exigua* individuals was recorded in 2020 (2,828 individuals), while the lowest was detected in 2012 (46 individuals). The annual number of *S. exigua* showed an interannual fluctuation in biomass, with one to three occurrence peaks per year from 2012 to 2022. There were four occurrence peaks in 2019: late June (159 individuals), early August (231 individuals), early September (165 individuals), and late October (135 individuals). There were two occurrence peaks of *S. exigua* in 2013-2017 and 2021.

Migration peaks occurred in mid-September (137 individuals) and mid-October (116 individuals) in 2013; early September (296 individuals) and mid-October (275 individuals)

**Table 1** Migration periods and abundances of *Spodoptera exigua* trapped by sex pheromone traps in Shenyang, northern China, during 2012–2022.

| Year | Date of first capture[a] | Date of final capture[a] | Migration duration (days) | Date of peak catches (n)[b] | Total catch |
|------|--------------------------|--------------------------|---------------------------|-----------------------------|-------------|
| 2012 | 6 Jul. (1) | 28 Sep. (1) | 84 | 7 Sep. (36) | 46 |
| 2013 | 24 Jun. (2) | 3 Nov. (3) | 102 | 15 Sep. (137) | 820 |
| 2014 | 9-Jun. (1) | 13 Oct. (275) | 126 | 25 Aug. (296) | 1,576 |
| 2015 | 25 May (1) | 18 Oct. (79) | 146 | 10 Aug. (274) | 1,881 |
| 2016 | 23 May (1) | 21 Sep. (35) | 121 | 24 Aug. (105) | 614 |
| 2017 | 31 May (1) | 12 Oct. (17) | 134 | 10 Aug. (105) | 1,511 |
| 2018 | 28 May (2) | 19 Nov. (1) | 175 | 7 Aug. (388) | 2,002 |
| 2019 | 20 May (1) | 19 Nov. (1) | 183 | 12 Aug. (231) | 1,942 |
| 2020 | 9 Jun. (2) | 24 Nov. (10) | 168 | 11 Aug. (948) | 2,828 |
| 2021 | 28 May (1) | 5 Nov. (9) | 161 | 30 Jul. (98) | 497 |
| 2022 | 20 Jun. (8) | 7 Nov. (4) | 150 | 15 Aug. (81) | 378 |

Notes.
[a] Trapped amounts of *S. exigua* are given in parentheses.
[b] The trapped number of *S. exigua* is given in parentheses next to the name of the months.

in 2014; early August (274 individuals) and middle September (228 individuals) in 2015; middle August (105 individuals) and middle September (77 individuals) in 2016; early August (248 individuals) and early September (208 individuals) in 2017; and late July (98 individuals) and late August (91 individuals) in 2021. Only one peak occurred in early September 2012 (36 individuals), early August 2018 (388 individuals), early August 2020 (948 individuals) and early August 2022 (81 individuals) (Fig. S1).

## Interannual and seasonal migration patterns of *S. exigua*

From 2012 to 2022, *S. exigua* was initially captured in Northeast China between late May and late June, with the earliest occurrence recorded on 20 May 2019. The final appearance of *S. exigua* occurred on 24 November 2020. The longest and shortest occurrence periods were observed in 2019 (183 d) and 2012 (84 d), respectively, with an average occurrence period of 140.9 ± 9.34 d (Table 1). The cluster analysis showed that the total annual number of trapped *S. exigua* was divided into four categories (Fig. S2; Table S1). The largest occurrence occurred in 2020, with 2,828 catches; mass occurrence occurred in 2014, 2015 and 2017–2019; normal occurrence occurred in 2013, 2016, 2021 and 2012; and weak occurrence occurred in 2012, with only 46 catches. In addition, the results of the generalized linear mixed-effect Poisson regression demonstrated significant variations in the weekly captures across different months. Although a significant interaction between year and month was identified, the effect of year was not significant ($P = 0.061$) (Table 2).

From 2012 to 2022, the numbers of *S. exigua* captured were as follows: 46 (2012), 820 (2013), 1,576 (2014), 1,881 (2015), 614 (2016), 1,511 (2017), 2,002 (2018), 1,942 (2019), 2,828 (2020), 497 (2021), and 378 (2022). The lowest and highest numbers of *S. exigua* were trapped in May and September, respectively, accounting for 34.65% ± 6.81% and 0.11% ± 0.04% of the annual totals, respectively. There were significant differences in the capture numbers of *S. exigua* among years ($F_{10,273} = 3.947$, $P < 0.01$) and months

**Table 2** Two-way ANOVA analysis of the number of *Spodoptera exigua* trapped by sex pheromone traps in Shenyang from May to October 2012–2022.

| Source | Type III sum of squares | df | Mean squares | F-values | P |
|---|---|---|---|---|---|
| Month | 242978.489 | 10 | 24297.849 | 5.356 | 0.000 |
| Year | 571413.128 | 6 | 95235.521 | 20.993 | 0.000 |
| Month × Year | 480453.315 | 60 | 8007.555 | 1.765 | 0.002 |
| Error | 948114.750 | 209 | 4536.434 | | |
| Total | 2981651.000 | 286 | | | |

($F_{6,273} = 28.226$, $P < 0.01$). Furthermore, we observed a steady increase in the number of captured *S. exigua* from May to September, followed by a decline in November (nonlinear model, $y = -0.007x^2 + 0.232x - 0.200$, $R^2 = 0.814$, $P < 0.05$). Additionally, we found that the number of captured *S. exigua* exhibited a significant increase during 2012–2018, and significant decrease in other four years (linear model, $y = 224.570x - 4513$, $R^2 = 0.446$; $y = -702.300x + 1 \times 10^6$, $R^2 = 0.588$, $P < 0.05$) (Fig. 3).

Detrended correspondence analysis (DCA) was conducted using monitoring data from 2012–2022, and the number of LGA results was less than 3. Therefore, redundancy analysis (RDA) was used, and the results of the study showed that the total variance explained by the two axes of RDA was 91.07%, and the ranking effect was good, which indicated that meteorological factors had a strong ability to explain the capture of *S. exigua* by sex traps. The results showed that the total variance in the four meteorological factors explained 59.5% of the catch of *S. exigua* (Table S2). The explanatory strengths were wind speed, temperature, humidity and rainfall in descending order, and the effect of wind speed on *S. exigua* catch reached a significant level ($F = 18.3$, $P < 0.005$) (Table S3). In summary, wind speed had a greater influence on *S. exigua* catches than temperature, humidity and rainfall. In addition, *S. exigua* catches in 2012, 2013, 2015, 2016, 2017, 2018, 2019, 2021, and 2022 were positively correlated with humidity, temperature, and rainfall; negatively correlated with wind speed, positively correlated with rainfall and humidity; and negatively correlated with wind speed and temperature in 2014 and 2022 (Fig. 4).

## Migratory paths of *S. exigua*

Based on the amount of *S. exigua* trapped in Shenyang from 2012 to 2021, the migration behavior and biological characteristics of *S. exigua* during typical migration events from June to September were selected, and backwards trajectory analysis of the migration path at 12/24 h was carried out using the HYSPLIT model. The results indicated that the sources of this pest varied across different months in Shenyang. The major source location in June was Shandong (61.62%), the major source locations in July were Shandong (28.24%), Liaoning (25.29%) and Jilin (15.14%), the major source locations in August were Shandong (37.58%), North Korea (22.77%) and Liaoning (11.4%). The potential major sources in September were Jilin (21.11%), Heilongjiang (15.63%) and Shandong (14.37%) (Fig. 5).

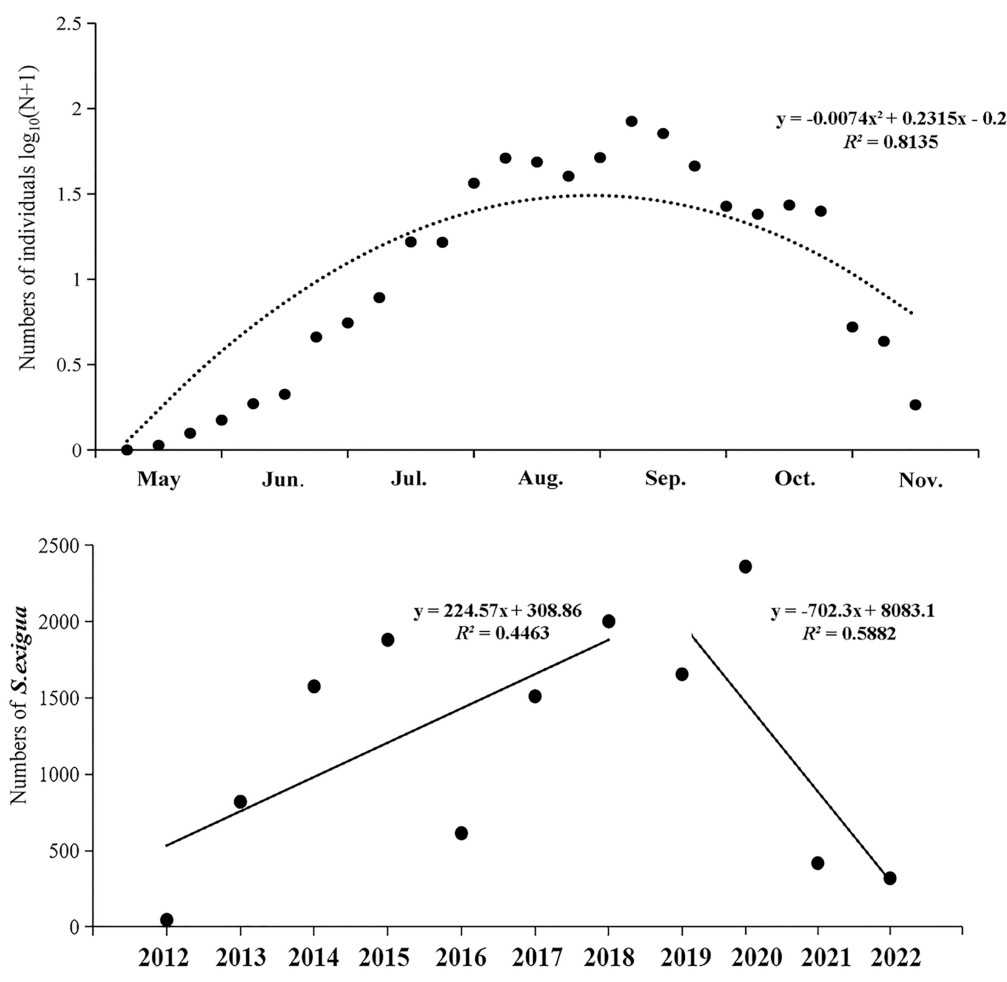

**Figure 3** **Seasonal captured numbers of *Spodoptera exigua* using sex pheromone traps in Shenyang, North China during 2012–2022.** The mean values of different variables are shown (±SE); Tukey's HSD test showed no significant difference at the 0.05 level between dots with the same letter.

## DISCUSSION

Migratory behaviour is the seasonal movement of populations back and forth between areas where survival and reproduction are either favourable or unfavourable (*Dingle & Drake, 2007*). The study of insect migration should be combined with meteorological conditions, especially when understanding the route and direction of insect migration (*Shamoun-Baranes, Bouten & van Loon, 2010*). Night-flying insects encounter wind systems that can span thousands of kilometers (*Burt & Pedgley, 1997*), which has been proven in *S. exigua* (*French, 1969*; *Mikkola, 1970*). It is assumed that this moth migration coincided with weather fronts, storms, or other meteorological events. Moreover, the number of moth catches exhibits significant variation across different months, a pattern that is consistent with observations in other agricultural pests, such as *Helicoverpa armigera* (*Feng et al., 2009*) and *Spodoptera litura* (*Fu et al., 2015*). We think that there are two reasons for the large annual fluctuations in biomass. One reason is that the number of migratory

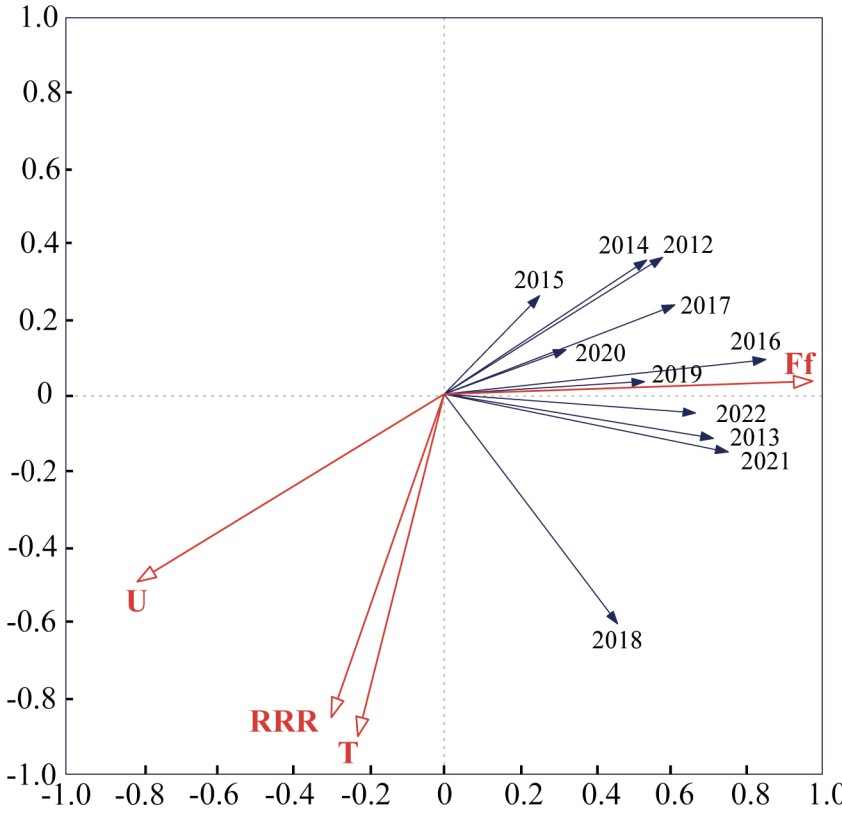

**Figure 4** **Redundancy analysis (RDA) of annual capture–meteorological factor correlations in Shenyang from 2012 to 2022.** The smaller the angle between the vectors (or a vector and an axis) and the longer the vectors are, the more correlated the variables represented by the vectors are. The abbreviations used are as follows: T: average weekly temperature; U: average weekly humidity; Ff: average weekly wind speed; and RRR: average weekly rainfall.

*S. exigua* in Shenyang varies greatly in different years. The other reason is that it has a direct relationship with the local population occurrence of *S. exigua*, which is related to the local climatic conditions (temperature, humidity, precipitation, *etc.*), host nutrition, natural enemies and other factors. Furthermore, studies have demonstrated significant variations in the occurrence of *S. exigua* biomass between different seasons and years, which was the first evidence that long-distance back-and-forth migration between overwintering areas and nonoverwintering areas is a frequent ecological phenomenon (*Fu et al., 2015*).

The occurrence biomass of *S. exigua* has been systematically monitored for many years using insect radar and searchlight traps. These monitoring methods have revealed that *S. exigua* can migrate across seas, and there are significant interannual and monthly differences in biomass (*Fu et al., 2017*). It has been observed that *S. exigua* regularly engages in sexual migration across Bohai Bay, and the number of *S. exigua* migrating southwards is significantly higher than the number migrating northwards (*Fu et al., 2017*). Similarly, in this study, the biomass in spring was significantly lower than that in summer and autumn. Specifically, the average total biomass of *S. exigua* in autumn was found to be more than 10 times higher than that observed in spring season. This pattern of migration

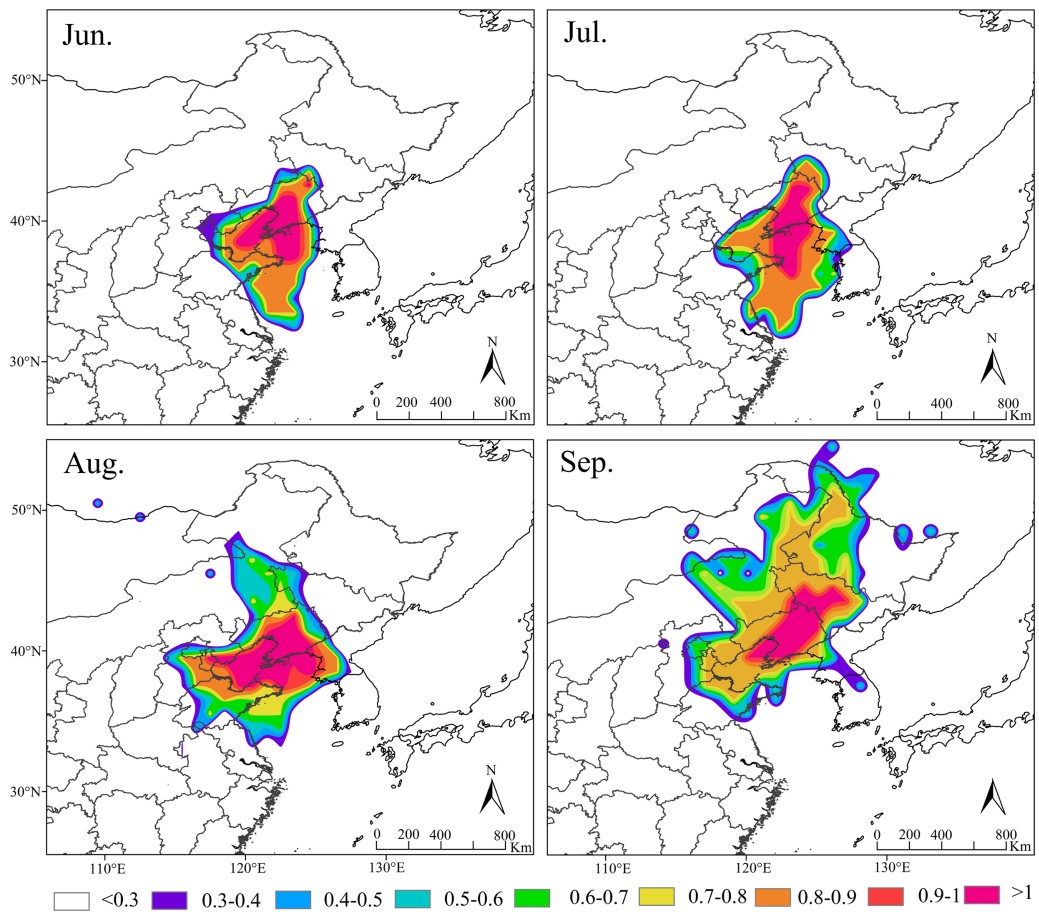

**Figure 5** **Seasonal migration trajectories originating from the source areas (represented by black lines) of *Spod optera exigua* trapped in Shenyang, along with their potential flight directions (illustrated by colored lines).** Note: The red star represents the location of Shenyang, Liaoning Province, China. Map credit: GADM database (http://www.gadm.org/) for free use; the figure is modified from the graphic of *Wang et al. (2023)*.

provides better reproductive benefits to the offspring population. In addition, radar observations and population dynamics monitoring of *S. exigua* in nonwintering areas have confirmed that *S. exigua* undergoes transregional migration across warm temperate and semiarid temperate climate zones (*Feng et al., 2003*; *Fu et al., 2017*). In addition, there was a significant increase in the biomass of *S. exigua* over the past 11 years. This increase is likely due to climate warming and the frequency of high temperatures in the southern China, which caused this moth pest to migrate northwards in search of suitable habitats and breeding grounds. All *S. exigua* were captured using sex pheromone traps between May and November, and the earliest capture of these adults took place on 20 May 2019, while the final capture was recorded on 24 November 2020. *S. exigua* cannot overwinter in Northeast China. Therefore, the adults captured in mid-June were determined to be immigrants, potentially originating from regions further south, such as eastern and central regions of China or even farther to the south.
Trajectory analysis is a powerful tool for tracing the origin and predicting populations of migratory insects (*Chapman et al., 2010*). The accuracy of trajectory analyses relies on setting the appropriate parameters for flight, such as time, speed, and altitude. An increase in flight altitude results in an increase in flight distance, and flight altitude also influences flight direction during various migration events (*Hu et al., 2021*). The long-distance migration of insects is primarily guided by the wind load of the seasonal upper air flow and is related to the seasonal change in wind pattern. In temperate regions, East Asian monsoon currents play a crucial role in providing favourable paths for long-distance insect migration (*Drake & Farrow, 1988*). Both pests and natural enemies migrate toward the south with the aid of winter winds as a means to escape harsh winters and to overwinter and reproduce in autumn. In fact, populations of *S. exigua* originate in southern perennially damaged and/or overwintering areas. These areas are characterized by the prevalence of monsoons during late spring and early summer. As a result, the migration pattern of *S. exigua* is ranged from south to north during this period. This migratory behavior is driven by the need to find suitable habitats and resources as the season changes. In northern China, *S. exigua* is becoming increasingly harmful to a variety of crops, at least in part because of its strong migratory ability (*Han et al., 2004*). In fact, *S. exigua* exhibits similar behavior to that of other species due to its inability to survive at high latitudes during winter (*Feng et al., 2003*; *Mikkola & Salmensuu, 1965*). The migration destination for *S. exigua* is either the wintering area or the perennially damaged area located in the southern region. For instance, in late September in Langfang city, Hebei Province, a searchlight trap was used concurrently with insect radar to observe the migratory routes of northern moths during the nighttime in northerly or northeasterly winds (*Feng et al., 2003*). Notably, the southeast monsoon, which predominantly occurs in South and Southeast Asia, typically flows from the eastern Pacific Ocean toward the southern regions (April), to the middle and lower reaches of the Yangtze River (late May to early July), and then to northern and northeastern regions (late July to early August) in China. Therefore, different migration routes of *S. exigua* are assumed to occur (1) from south to north in May–July; (2) from the south to the center in May and from the center to the north in July; and (3) from central migration to the north in July. In general, many species, such as *Danaus plexippus*, have seasonal movements (*Froy et al., 2003*). Thus, *S. exigua* continues to breed in different parts of Southeast Asia and South Asia, and understanding its migration patterns and potential climate determinants can facilitate pest prediction, predict the emergence of pesticide resistance, and aid prevention and control (*Rochester et al., 1996*; *Carrière, Crowder & Tabashnik, 2010*; *Wu et al., 2020*).

In general, *S. exigua* breeds continuously in areas where the mean January temperature exceeds 12 °C and overwinters between January 4–12 °C isotherms (*Jiang & Luo, 2010*). Based upon the local climatic conditions, different areas can thus be delineated for *S. exigua* overwintering, year-round breeding or non-overwintering (*Jiang & Luo, 2010*). Transregional migration across temperate and semiarid temperate zones permits the annually recurring colonization of nonoverwintering areas (*Feng et al., 2003*; *Fu et al., 2017*). *S. exigua* cannot overwinter in Northeast China. The populations captured by sex pheromones in May and June are usually migratory populations, and in September, they are migratory populations that move back to

the southern overwintering area. Our previous studies focused mainly on the genetic differentiation and gene flow among different geographical populations of *S. exigua*. Furthermore, we discovered that the annual number of captured *S. exigua* was correlated with the numbers of Beihuang Island (BH) and Shenyang (SY) ($F_{1,46} = 79.33$, $P < 0.0001$), and the population dynamics of *S. exigua* were similar between both sites from 2012 to 2019 (*Wang et al., 2022*). Therefore, sex pheromone monitoring can initially reflect the occurrence or migration period of *S. exigua*. In contrast, in this study, we focused mainly on the population dynamics and seasonal migration patterns of *S. exigua* based on 11 years of monitoring data from 2012 to 2022. Moreover, we analysed the meteorological factors that affect insect migration and migration trajectories. Overall, *S. exigua* tended to migrate more from south to north in summer and from north to south in autumn. As a migratory pest, wind speed strongly affects the capture of *S. exigua*, which is detrimental to its ability to control sex pheromones. Therefore, this study holds great significance in terms of determining the migration path and source areas of immigrants and monitoring and providing early warnings for moth pests. In the future, we will attempt to accurately monitor the occurrence dynamics of *S. exigua* by using other monitoring methods, such as searchlight traps and lamp traps. In addition, insect radar will be used in combination with other technologies to extract accurate data on the high-altitude flight behavior parameters of target insects. In addition, insect migration is closely associated with meteorological factors. For example, temperature not only affects the number of aphids that migrate but also directly affects the duration of aphid migration (*Zhang, Xu & Tang, 1997*; *Cao et al., 2006*). When there is no wind or when wind speeds are low, migratory insects rarely or never take off (*Zhai & Zhang, 1993*). Similarly, in this study, the effect of wind speed on the capture of *S. exigua* was greater than that of temperature, humidity and rainfall.

Currently, chemical pesticides are the main strategy for controlling this moth in China (*Cho et al., 2018*). To optimize the use of pesticides, it is necessary to have a better understanding of insect population dynamics and economic thresholds. Due to long-term exposure to insecticides, this moth species has developed resistance to a variety of chemical insecticides. There are more than 86 incidences of pesticide resistance among *S. exigua* populations worldwide. Moreover, resistance to new chemical pesticides has also emerged in several Asian countries (*Ahmad, Farid & Saeed, 2018*; *Cho et al., 2018*). The large-scale migration of pests can result in the rapid spread of resistant insects. Therefore, the objective of this study was to investigate the population dynamics and migration patterns of this moth pest. The ultimate goal was to develop early predictions for major migratory pests and provide a scientific foundation for integrated pest management strategies. Similarly, by linking the migration patterns of *S. exigua* to the Asian monsoon circulation system, researchers can gain insights into climate-driven ecological disruptions, range shifts, or socioeconomic impacts (*Zeng et al., 2020*). By elucidating the migration dynamics of *S. exigua* driven by monsoons, we can predict the number, duration and geographical pattern of migratory populations of beet moths in China. Furthermore, in the future, pesticide resistance monitoring, population genetics, isotope tracing analysis could be utilized to obtain more precise migration trajectories of *S. exigua*. Further investigations also are necessary to study the timing and distance of migration, sex ratio, mating frequency, and

ovarian development, *etc.* Expanding and strengthening monitoring networks will not provide early warnings of *S. exigua* outbreaks but also yield valuable data for effective control of other migratory insects. To carry out pest forecasting work and effectively monitor the daily population dynamics of *S. exigua*, we will employ a combination of searchlight traps, lamp traps, and field surveys. Therefore, our work helps to elucidate the large spatial scale population dynamics of *S. exigua* and provides important information for its monitoring, prediction and IPM methods.

## CONCLUSIONS

This study provides further data on the regularity of population occurrence and seasonal migration patterns of *S. exigua* in northern China. The results support the significant interannual and seasonal differences in the capture numbers of *S. exigua*, as well as in the diverse source regions in different months, which primarily originated from Northeast China and East China. These unique insights into migration patterns will help increase the understanding of the occurrence of the beet armyworm in China and provide an important basis for regional prevention and control.

## ACKNOWLEDGEMENTS

We are grateful to Associate Research Fellow, Bin Lv, at the Chengdu Institute of Biology, Chinese Academy of Sciences, Chengdu, China, for providing valuable comments and suggestions. We are also grateful to many researchers forproviding some enthusiastic help during our field surveys.

### Funding

This research was supported by the Natural Science Foundation of Liaoning Province of China (No. 2023-MS-209); The National Key R & D Program of China (2021YFD1400200) and the National Natural Science Foundation of China (No. 31871950). The funders had no role in study design, data collection and analysis, decision to publish, or preparation of the manuscript.

### Grant Disclosures

The following grant information was disclosed by the authors:
The Natural Science Foundation of Liaoning Province of China: No. 2023-MS-209.
The National Key R & D Program of China: 2021YFD1400200.
The National Natural Science Foundation of China: No. 31871950.

### Competing Interests

The authors declare there are no competing interests.

## Author Contributions

- Hao-Tian Ma conceived and designed the experiments, prepared figures and/or tables, and approved the final draft.
- Li-Hong Zhou conceived and designed the experiments, performed the experiments, prepared figures and/or tables, authored or reviewed drafts of the article, and approved the final draft.
- Hao Tan performed the experiments, authored or reviewed drafts of the article, and approved the final draft.
- Xian-Zhi Xiu analyzed the data, authored or reviewed drafts of the article, and approved the final draft.
- Jin-Yang Wang analyzed the data, authored or reviewed drafts of the article, and approved the final draft.
- Xing-Ya Wang analyzed the data, authored or reviewed drafts of the article, and approved the final draft.

## Field Study Permissions

The following information was supplied relating to field study approvals (i.e., approving body and any reference numbers):

Field experiments were approved by the Chinese Academy of Agricultural Sciences (project number: 201003025).

## Data Availability

The data are available at Zenodo: Hao-Tian Ma, Li-Hong Zhou, Hao Tan, Xian-Zhi Xiu, Jin-Yang Wang, & Xing-Ya Wang (2023). Population dynamics and seasonal migration patterns of *Spodoptera exigua* in northern China based on 11 years of monitoring data (raw data) [data set]. Zenodo. https://doi.org/10.5281/zenodo.8275098.

## Supplemental Information

Supplemental information for this article can be found online at http://dx.doi.org/10.7717/peerj.17223#supplemental-information.

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
