# Peer review of "Population dynamics and seasonal migration patterns of Spodoptera exigua in northern China based on 11 years of monitoring data"

_PeerJ, doi:10.7717/peerj.17223_

## Round 0.1 · original submission · Major Revisions

Please, take into account all reviewers' comments.

**Language Note:** The review process has identified that the English language must be improved. PeerJ can provide language editing services - please contact us at copyediting@peerj.com for pricing (be sure to provide your manuscript number and title). Alternatively, you should make your own arrangements to improve the language quality and provide details in your response letter. – PeerJ Staff

Reviewer 1 ·

Basic reporting

The authors have analyzed the beet armyworm's population dynamics and seasonal migration. The research topic is interesting and novel; there are not many studies analyzing the migration pattern of migratory lepidopterans. However, I have some suggestions that the authors should consider to improve the manuscript quality further. Specifically, the Introduction requires some additional background information on insect migration to strengthen the novelty of this study.

First, I think the authors should provide more information on insect or Lepidoptera migration, especially how understudied the topic is compared to vertebrates. For example, to improve novelty, the authors could include one/two sentence/s saying that while 3.29 % of migratory butterflies are migratory (https://doi.org/10.1111/brv.12714), most studies are on a few species, and the status and population dynamics of the majority of species remain unknown (https://doi.org/10.1002/inc3.13).

Lines 80-83: There are a few other methods as well. Please see this recent review: https://doi.org/10.1002/inc3.13.

Line 148: Remove ‘_’

The method section is well-described. However, I think a methodological figure would help the readers to understand and repeat the study.

Something strange happened in the Figure captions. Figure 1 is missing! I think Figure 2 is posted as

Figure and the same is true for the rest of the Figures.

Figure 1 (as written): What are these box plots?

Line 303: This is a more accurate citation: https://doi.org/10.1073/pnas.2102762118.

Experimental design

no comment

Validity of the findings

no comment

Additional comments

no comment

Reviewer 2 ·

Basic reporting

English language needs editing by a fluent speaker for the clarity. Authors need to write the document coherently, one of the major issue I noted at several places literature appended was misinterpreted. Furthermore, few references presented information that was lacking in the source cited.

Experimental design

Materials and methods section need more details for trapping of the insects ( e.g. trap size, structure, interval after which pheromone was replenished and chemical information etc.) Reference Lin (1963) to determine peak of migration events was lacking in the literature cited.

Validity of the findings

Implications of the study are lacking i.e. how will results of current study move the pest management of S. exigua forward ? Needs to clearly state the novelty, as the result of migration and interannual differences from the same region have been published (Wang et al., 2022). Conclusions drawn have been superficially interpreted.

Additional comments

All the document needs to be rewritten coherently and logically with in-depth digestion of the literature cited.

---

## Round 0.2 · accepted · Accept

Thank you for addressing all of the reviewers' comments. In my opinion, the manuscript is now ready for publication.

Reviewer 1 ·

Basic reporting

The authors have substantially revised the manuscript. I am satisfied with all the changes and I think the manuscript can be accepted. I look forward to reading the published version.

Experimental design

NA

Validity of the findings

NA

Additional comments

NA